# Relationships within *Bolbitis sinensis* Species Complex Using RAD Sequencing

**DOI:** 10.3390/plants13141987

**Published:** 2024-07-20

**Authors:** Liyun Nie, Yuhan Fang, Zengqiang Xia, Xueying Wei, Zhiqiang Wu, Yuehong Yan, Faguo Wang

**Affiliations:** 1Guangdong Provincial Key Laboratory of Applied Botany, South China Botanical Garden, Chinese Academy of Sciences, Guangzhou 510650, China; nieliyun18@163.com (L.N.); fangyuhan@scbg.ac.cn (Y.F.); xiazengqiang@scbg.sc.cn (Z.X.); hsdweixueying@163.com (X.W.); 2Shenzhen Branch, Guangdong Laboratory for Lingnan Modern Agriculture, Genome Analysis Laboratory of the Ministry of Agriculture, Agricultural Genomics Institute at Shenzhen, Chinese Academy of Agricultural Sciences, Shenzhen 518120, China; wuzhiqiang@caas.cn; 3School of Medical, Molecular and Forensic Sciences, Murdoch University, Murdoch, WA 6149, Australia; 4University of Chinese Academy of Sciences, Beijing 100049, China; 5Shenzhen Key Laboratory for Orchid Conservation and Utilization, The National Orchid Conservation Center of China and the Orchid Conservation & Research Center of Shenzhen, Shenzhen 518114, China; yhyan@sibs.ac.cn

**Keywords:** *Bolbitis*, species delimitation, phylogenetic analysis, genetic diversity and differentiation

## Abstract

Species identification and phylogenetic relationship clarification are fundamental goals in species delimitation. However, these tasks pose challenges when based on morphologies, geographic distribution, and genomic data. Previously, two species of the fern genus *Bolbitis*, *B.* × *multipinna* and *B. longiaurita* were described based on morphological traits; they are phylogenetically intertwined with *B. sinensis* and fail to form monophyletic groups. To address the unclear phylogenetic relationships within the *B. sinensis* species complex, RAD sequencing was performed on 65 individuals from five populations. Our integrated analysis of phylogenetic trees, neighbor nets, and genetic structures indicate that the *B. sinensis* species complex should not be considered as separate species. Moreover, our findings reveal differences in the degree of genetic differentiation among the five populations, ranging from low to moderate, which might be influenced by geographical distance and gene flow. The Fst values also confirmed that genetic differentiation intensifies with increasing geographic distance. Collectively, this study clarifies the complex phylogenetic relationships within the *B. sinensis* species complex, elucidates the genetic diversity and differentiation across the studied populations, and offers valuable genetic insights that contribute to the broader study of evolutionary relationships and population genetics within the *Bolbitis* species.

## 1. Introduction

The identification, delimitation, and description of species have long been subjects of intense debate in the fields of systematic and evolutionary biology [1,2]. Species delimitation involves determining which groups of individual organisms represent distinct populations within a single species and which represent separate species [3]. Historically, taxonomists described new species based on morphological characteristics. However, the variation in certain morphologies in response to environmental factors or evolutionary processes has often led to confusion in classification [2,4]. In recent decades, a proliferation of methods utilizing molecular data has emerged to propose species hypotheses [5]. These advances have provided evidence of natural hybridization, reticulate evolution, adaptive radiation, and extensive gene flow after speciation, revealing a complex and often intricate history of species. Concurrently, these advances have also necessitated a reconsideration of the concept of “species” [6,7].

*Bolbitis* Schott is a pantropical fern genus within the family Dryopteridaceae, comprising approximately 80 species primarily found in tropical Asia [8,9]. Species of *Bolbitis* are typically terrestrial, lithophytic, or occasionally epiphytic on tree trunks, featuring creeping or shortly erect rhizomes, and are commonly found in damp forests (Figure 1). *Bolbitis* is characterized by dimorphic fronds and proliferous adaxial buds on the apex of terminal segments [10]. *Bolbitis* × *multipinna* is endemic to China and is solely distributed in Yunnan Province [11,12]. Due to its irregular venation pattern, with one to more anastomosing veinlets arising from its lateral veins, *B.* × *multipinna* has been described as an inter-specific hybrid of *B. angustipinna* × *B. sinensis* [13,14]. Another species identified in 2006, *Bolbitis longiaurita*, bears similarities to *B. sinensis*, but differs in having an unwinged rhachis, as well as in the base of its two lowermost sterile pinnae being asymmetrical, with the two or three basiscopic lobes undeveloped and the other lobes longer than the acroscopic ones [15]. However, according to field surveys, we found that *B.* × *multipinna* and *B. longiaurita* do not exhibit independent geographical distributions and are consistently found sympatrically with *B. sinensis*. Another noteworthy observation is the lack, or instability, of morphological characteristics that clearly distinguish these three species (Figure 1). One available characteristic that distinguishes *B. sinensis* and *B.* × *multipinna* is the presence of one to multiple anastomosing veinlets. Similarly, *B. longiaurita* displayed only one unstable pinnae character when compared to *B. sinensis* (i.e., basally elongated lobes). The latest phylogeny, based on three chloroplast sequences, resolved *Bolbitis* into four clades: the Malagasy/Mascarene clade, the African clade, the American clade, and the Asian clade. *B. sinensis*, *B.* × *multipinna* and *B. longiaurita* formed a well-supported group in the Asian clade. The most recent common ancestor (MRCA) of the *B. sinensis* species complex likely originated in subtropical to tropical Asia and diverged around 3.84 Mya [16]. However, the species within the *B. sinensis* complex are nested within each other, and the species diversity remains uncertain. Therefore, more extensive sampling is required to elucidate the reticulate relationships within the *B. sinensis* species complex.

Genomic data contains extensive information about the degree of genetic isolation among species, as well as ancient and recent introgression events. Consequently, genomic data can play an important role in species delimitation across various species concepts [3,17]. High-throughput techniques based on restriction site-associated DNA sequencing (RAD-seq) are enabling the low-cost discovery and genotyping of thousands of genetic markers for species, especially for non-model and non-genome species [17,18,19,20]. They have been widely used to study genomic evolution, especially at intergeneric, interspecific, and intraspecific taxonomic levels, including for Amazonian bryophytes [21], global oaks [22,23], Arundinarieae of Poaceae [24,25], the grape genus [26], *Cunninghamia lanceolata* in the Cupressaceae family [27], and the allopolyploid tree fern *Gymnosphaera metteniana* [28].

In this study, we performed RAD sequencing on 65 individuals from five populations of the *B. sinensis* species complex, which includes *B. sinensis*, *B.* × *multipinna* and *B. longiaurita*. The population-based data were used to: (1) reconstruct the phylogenetic relationships within the *B. sinensis* species complex based on genomic variations and (2) identify genetic diversity and genetic differentiation within the species complex.

## 2. Results

### 2.1. Sample Collection and SNP Calling

To investigate the *B. sinensis* species complex, we collected related species from Yunnan, China. A total of 65 individuals representing five populations of the *B. sinensis* complex were collected (Table 1 and Appendix A), including 26 *B. sinensis*, 31 *B.* × *multipinna*, and four *B. longiaurita*. Additionally, four individuals could not be identified to the species level and were designated as *B.* sp. All samples underwent RAD sequencing. After quality filtering, a total of 840 Gb clean data were obtained from 65 individuals by sequencing RAD libraries, with an average of 13.67 Gb pair-end clean data per individual (Appendix A). The retained reads were finally assembled into an average of 182,965 RAD stacks per individual using *ustacks*, with a mean coverage depth of 25.41× (Appendix A). A catalog of 8,485,326 putative loci was constructed using *cstacks*, and an average of 110,868 putative loci per individual was matched to the catalog using *sstacks*. A total of 92% of read pairs were identified as putative PCR duplicates and removed by *gstacks*. Finally, 8,392,168 loci were genotyped, with an effective per-sample mean coverage of 1.7×. In the *population* analysis, we applied five -p and seven -r parameters to filter SNPs and obtained 35 SNP datasets, retaining variant sites ranging from 89 in the p5-r0.90 dataset to 30,654 in the p1-r0.30 dataset (Appendix A).

### 2.2. Phylogenetic Trees and Neighbor Net

To identify consensus and differences among a large set of trees (Appendix A), a density tree was visualized for comparison (Figure 2A). The phylogenetic density tree revealed that, despite some inconsistency in the topological structures derived from different datasets, there was a consistent pattern in which the BB and NG/NP population samples formed a group, while the PT and the remaining NG/NP/ML samples formed another group. The three species within the *B. sinensis* species complex did not form monophyletic branches or clades but displayed consistent geographical differentiation.

Additionally, to compare topologies among trees, Robinson–Foulds (RF) distances between a set of trees were computed and analyzed (Appendix A). The results showed that the p1 dataset had a relative lower RF distance than other datasets (Figure 3A and Appendix A). The p1-r0.60 dataset was selected as a representative dataset for visualization based on the high support value of major branches and low RF distance. The phylogenetic tree from the p1-r0.60 dataset (Figure 2B) revealed that BB and PT populations had high support rates (MLBS > 90%) and relatively stable positions, while the NP/NG populations formed three branches, with one branch having a low support rate (MLBS < 70%). Group BB comprised 18 individuals of *B. sinensis* and *B. × multipinna* from Bubeng, Yunnan, China (MLBS = 98%). Similarly, Group ML included six individuals of *B. sinensis* and *B. × multipinna* from Menglun, Yunnan, China (MLBS = 98%), along with one *B. longiaurita* sample from the NP population. Group PT contained 16 individuals of *B. sinensis*, *B. × multipinna*, and *B.* sp. from Puteng, Yunnan, China (MLBS = 100%), along with two additional *B. sinensis* individuals from the BB population and one *B. × multipinna* from the NG population. However, individuals from the NP and NG populations largely intermixed and further divided into three branches, showing a genetic distance in between Group BB and Group PT. Based on the p1-r0.60 dataset, we also generated a phylogenetic network in SplitsTree4 to visualize the relationships between populations (Figure 2C). The neighbor net split graphs showed some conflict signals among the five populations. Major box-like splits were detected within the BB and PT populations, which indicates a certain amount of topological ambiguity.

### 2.3. Genetic Structure among B. sinensis Species Complex

To further explore the relationship among *B. sinensis* population groups and examine the population substructure, structure and PCA analysis were performed based on 35 SNP datasets. The cross validation (CV) error analysis showed that a total of 22 datasets including the p1-r0.60 dataset supported one genetic group (K = 1) as optimal for *B. sinensis* population groups (Figure 3A, Appendix A). In addition, nine datasets containing a small number of core SNPs with a low missing rate across all populations indicated that K = 2 was the optimal value. The remaining four datasets supported K values of 3 or 4. Among the K = 2 datasets, the p3-r0.70 dataset (1031 SNPs with a low missing rate) with the highest ML tree support values and a low RF distance was selected for comparison (Figure 3B). This analysis showed that the vast majority of individuals shared genetic components with either the BB population or the PT population. Two mixed patterns were observed in the NG, NP, and ML populations: one with the same genetic component as PT individuals and the other with polymorphic components originating from both the BB and PT populations. Within the BB population, individuals 110905-6, 110905-7, and 110905-8 showed the same genetic component as the PT population (Appendix A), which was largely consistent with the results of ML tree results (110905-7 and 110905-8 in Group PT, 110905-6 in Group NG+NP).

The two SNP datasets used above, p1-r0.60 and p1-r0.70, represented two different SNP strategies: a large number of SNPs with a high missing rate and a small number of SNPs with a low missing value, respectively. Both datasets were applied to PCA analysis (Figure 3C). The first two components of the p1-r0.60 dataset explained 8.41% and 3.23% of the total variance, while the p3-r0.70 dataset explained 12.30% and 4.75% of the total variance. Both analyses clearly divided the five populations of the *B. sinensis* species complex into three clusters: BB and PT were relatively independent and distant, while NG, NP, and ML formed a mixed cluster between BB and PT.

### 2.4. Genetic Diversity and Differentiation among Five Populations

To determine the genetic diversity among the five populations, genetic diversity indices for each population were calculated based on the p1-r0.60 SNP dataset (Table 2 and Appendix A). All populations exhibited private alleles, with the PT population showing the highest number of private alleles, followed by the NG population. The NP population had the lowest number of private alleles. In terms of the nucleotide diversity parameter (Pi), the ML population displayed the highest nucleotide diversity (Pi = 0.200), followed by PT (Pi = 0.158), NP (Pi = 0.156), and NG (Pi = 0.155). The BB population had the lowest nucleotide diversity (Pi = 0.115). Observed and expected heterozygosity of the five populations ranged from Obs_Het = 0.117 to 0.222 and Exp_Het  =  0.112 to 0.183, respectively. The population in BB had the lowest observed and expected heterozygosity (Obs_Het  =  0.116 and Exp_Het  =  0.112). The mean heterozygosities of the NG population (Obs_Het  =  0.160 and Exp_Het  =  0.147) were similar to those of the NP population (Obs_Het = 0.163 and Exp_Het = 0.142) and the PT population (Obs_Het = 0.169 and Exp_Het  =  0.150). Regarding the inbreeding coefficient (Fis), the p1-r0.60 dataset generated Fis values ranging from −0.015 to 0.03708. The genetic differentiation index (Fst) was also computed to evaluate the degree of differentiation among the five populations (Table 3). The highest Fst was observed between the BB and PT populations (Fst = 0.1574). The lowest Fst values were found between the NG and NP populations (Fst  ≈  0), followed by NP and ML (Fst  =  0.0198) and NG and ML (Fst  =  0.0258). These results indicated different degrees of genetic differentiation between geographically distinct populations. The high relative genetic diversity and low genetic differentiation suggested the existence of gene flow among specific populations.

### 2.5. Potential Gene Flow among Five Populations

To investigate whether tree discordances and reticulate networks were due to past introgression events, we calculated D statistics for all possible trios of populations using PT—which had the most independent genetic component—as an outgroup. Among all trios, 49 trios (out of 165 tested) showed significant D statistic values exceeding 0.1 after correcting for multiple testing (Appendix A and Appendix A and Figure 4). These significant trios were mostly found between Mul_BB group and the remaining subpopulations. Additionally, we calculated f4 ratios to estimate the amount of ancestry in an admixed population that came from potential donor populations. We found that only 19 trios exhibited significant excess allele sharing and ancestry proportions (f4-ratio) over 0.3. Consistently, most of these significant trios were found between the Mul_BB group and the remaining subpopulations.

## 3. Discussion

### 3.1. Reclassification of Bolbitis sinensis Species Complex

Identifying species and clarifying their phylogenetic relationships are fundamental goals of species delimitation, but this process is often challenging and controversial [2]. Previously, two species, *B. × multipinna* and *B. longiaurita*, were published primarily based on their morphological characters [11,12,15]. Additionally, *B. × multipinna* was thought to be a hybrid species in earlier research [13]. However, we found that the key taxonomic traits, such as venation patterns, seem to be unstable, which is considered a critical trait of *Bolbitis* [13,14,15]. In previous phylogenetic studies, the *B. sinensis* species complex formed a well-supported group within the Heteroclitae subclade. The MRCA of the *B. sinensis* species complex likely originated in subtropical to tropical Asia and diverged around 3.84 Mya [16]. Notably, within the Heteroclitae subclade, the *B. sinensis* species complex and another two species (*B. major* and *B. tonkinensis*) with free venation in common are not resolved as monophyletic, whereas other species with anastomosing veins are monophyletic [16]. Hence, this raises several questions: Do the variable individuals in the *B. sinensis* species complex belong to one species or to different species? Do the intermediate phenotypes truly suggest a reticulate evolution or a hybridization origin in *Bolbitis*? These questions highlight the complexity and need for further investigation into the species boundaries and evolutionary history within this group.

According to the phylogenetic tree, neighbor net, and genetic structure analysis, our results supported that the *B. sinensis* species complex cannot be regarded as separated species. There is notable genetic differentiation and gene exchange among specific populations. Combining these findings with previous studies, there is no clear evidence suggesting a hybrid origin for *B. × multipinna*. Regarding the morphological variation observed in the *B. sinensis* species complex, besides the stochastic nature of the trait itself, we speculate that gene flow between the *B. sinensis* species complex and other species may contribute to this variation.

Another possible reason for morphological variation lies in the mode of reproduction. Notably, one of the distinguishing features of *Bolbitis* is its ability to reproduce asexually via buds on leaves, which allows for rapid and efficient population expansion. However, for species with large and complex genomes, maintaining DNA replication fidelity through asexual reproduction can be challenging. Several studies have proposed that asexual populations have weaker responses to natural selection, leading to the accumulation of more deleterious mutations over generations [29,30,31]. The relatively high frequencies of asexual reproduction without ploidy cycling in ferns might be a consequence of frequent polyploidization, which allows for the buffering of recessive deleterious mutations over long time periods [31]. However, this remains a hypothesis, and more studies are needed to verify these observations and better understand the genetic and evolutionary dynamics within the *B. sinensis* species complex.

### 3.2. Genetic Diversity and Differentiation among Five Populations

The distribution of genetic diversity across populations is strongly determined by dispersal and gene flow patterns [32]. Two main patterns of gene flow are reported to be associated with population divergence: isolation-by-distance (IBD) and isolation-by-environment (IBE) [33,34,35]. In this study, different degrees of genetic diversity and differentiation were identified among the five populations. Higher genetic diversity was identified among the NP, NG, ML, and PT populations (Pi = 0.15601–0.2000), whereas the BB population exhibited the lowest nucleotide diversity (Pi = 0.11527). The geographical distance is shortest between the BB and NP populations, followed by BB and NG, BB and ML, and BB and PT. Three populations (NP, NG, and ML) are very close to each other geographically. Consistently, the Fst value between the BB and NP populations was the lowest (Fst = 0.0689), followed by the BB–NG and BB–ML pairs (Fst = 0.0710 and 0.0834, respectively). In contrast, the BB–PT pair had the highest Fst value (Fst = 0.1574). The lowest Fst values were found between NG and NP (Fst ~ 0), followed by NP and ML (Fst  =  0.0198) and NG and ML (Fst  =  0.0258). According to previous established criteria [36], five populations of the *B. sinensis* species complex showed low to moderate differentiation from each other (FST < 0.05 indicates low genetic differentiation, 0.05 < FST < 0.15 indicates moderate genetic differentiation). Moreover, it was also suggested that genetic differentiation between *B. sinensis* species complex populations increases with geographical distance. Previously, a large number of studies proposed a positive correlation between genetic and geographic distances in plants, such as *Mimulus guttatus*, *Paeonia decomposita*, and the tertiary relict species *Emmenopterys henryi* [37,38,39]. Even for a cosmopolitan marine planktonic diatom, a strong isolation by distance pattern has been identified at a large geographical scale [40]. For short-distance dispersal, Ledent et al. [21] reported significant isolation-by-distance (IBD) patterns in Amazonian bryophytes, where spatial genetic structures diminished beyond the limits of short-distance dispersal. Similar to bryophytes, populations with intermediate phenotypes within the *B. sinensis* species complex have a very restricted geographic distribution. This suggests that limited dispersal due to geographic distance may be a crucial factor in the genetic differentiation between BB and PT populations.

In addition, direct gene flow might also influence population structures and result in variation in genetic diversity. The low genetic differentiation and high genetic diversity within the NP/NG/ML population suggest ongoing gene flow among these populations. Our gene flow analysis identified significant signals of gene flow between the three sympatric populations and the BB population. For instance, the Patterson’s D statistic (D = 0.238821) and the f4 ratio (f4 ratio = 0.379994) of (Lon_NG, Mul_BB), Sin_ML), PT) trios test suggest that the Mul_BB and Sin_ML may share more derived alleles than the Lon_NG and Mul_BB populations. More trios with gene flow signals were detected in the BB population, indicating an asymmetric gene flow from BB to other populations. We speculated that this may be related to asymmetrical migration and high migration rates from BB to NP/NG/ML populations. Previous studies have found evidence, mainly at the inter-specific level, of asymmetric gene flow, such as asymmetrical genomic contributions of two parent species to homoploid hybrid *Ostryopsis* species [41,42]. Such asymmetries in hybridization might be related to the mating system/s of the species pair and other evolutionary forces [43]. Asymmetric gene flow between populations has progressively become the focus in landscape genetics [44]. A recent prospective study proposed that genetic differentiation, asymmetric gene flow, and genetic diversity in many tree species were shaped by global wind patterns [44]. Recently, Chang et al. [45] inferred four genetic clusters in wild barley (*Hordeum vulgare* L. ssp. *spontaneum*) populations of the Southern Levant using genotyping by sequencing (GBS) data. They also detected similar asymmetric gene flows among populations and found trends of gene flow in opposite directions in eastern and western regions. However, the evidence to support the asymmetric gene flow was not sufficient in this study. The results may also be influenced by the sampling size in each population. To clearly elucidate the gene flow events within the *B. sinensis* species complex, larger sample sizes, more population groups, and further experimental studies are needed in the future.

## 4. Materials and Methods

### 4.1. Taxon Sampling

In this study, we collected a total of 65 individuals representing five populations of the *B. sinensis* complex (Table 1). Species identification followed the criteria established by Dong and Zhang [14]. Specifically, 26 individuals were identified as *B. sinensis*, 31 individuals were identified as *B. × multipinna*, and four individuals were identified as *B. longiaurita*. Additionally, four individuals could not be identified to the species level and were designated as *B.* sp. Plant material was collected from living specimens in the field and stored in silica gel. The samples were carefully identified based on the descriptions in Flora of China, all specimens were deposited in the South China Botanical Garden (Guangzhou, China), and voucher numbers are given in Appendix A.

### 4.2. RAD-Seq Library Preparation and Sequencing

High-quality genomic DNA was extracted from leaf tissues using a modified CTAB procedure [46]. Genomic DNA was individually barcoded and processed into a reduced complexity library on the basis of the traditional single-digest RAD protocol described in the study of Ali, et al. [47], with the following process: (1) digesting genomic DNA from each accession with t*EcoRI* restriction enzyme, (2) ligating the digested product to a Solexa P1 adapter containing a 6 bp unique barcode, (3) pooling adapter-ligated fragments, (4) ligating a Solexa P2 adapter onto the ends of DNA fragments, (5) size selecting 200–400 bp fragments on agarose gels, (6) constructing individual libraries for each accession, and (7) enriching libraries with high-fidelity PCR amplification. Subsequently, paired-end sequencing was performed on an Illumina Hi-Seq X-Ten platform at Novogene Bioinformatics Institute (Beijing, China).

### 4.3. Data Processing

Raw data were trimmed for adapters and quality filtered before SNP calling. The quality of sequencing data was checked with FASTQC v0.11.9 (https://www.bioinformatics.babraham.ac.uk/projects/fastqc/ (accessed on 21 September 2022)) and visualized with MultiQC v1.13 (https://github.com/ewels/MultiQC (accessed on 21 September 2022)). Raw reads were filtered using Trimmomatic v0.39 with the parameters HEADCROP:15 and AVGQUAL:30 [48]. Then, the STACKS de novo pipeline [49,50] (http://catchenlab.life.illinois.edu/stacks/ (accessed on 28 September 2022)) was used to construct putative loci from short-read sequences and perform SNP calling. For each sample, the *ustacks* program in STACKS was executed to assemble identical sequences into putative alleles (primary stacks). The minimum sequence depth parameter, m, was set to five (m = 5), and all sequences had a nuclear distance less than or equal to the chosen value of the M parameter (M = 2). The complete catalog of loci from all individuals was created using *cstacks* program with a maximum between-individual distance parameter of two (n = 2). Following the cstacks catalog building, the *sstacks* program was used to identify the matching catalog locus for each of the de novo loci in each individual. Considering that a large percentage of duplicate sequences exist in RAD sequencing data, we use --rm-pcr-duplicates in *gstacks* program to minimize the effect of duplicate sequences. Finally, *populations* program were implemented to identify SNPs with the following parameter settings: (a) minimum number of populations a locus must be present in to process a locus: -p = 1, 2, 3, 4, 5; (b) minimum percentage of individuals in a population required to process a locus for that population: -r = 0.30, 0.40, 0.50, 0.60, 0.70, 0.80, 0.90; (c) a minimum minor allele frequency required to process a nucleotide site at a locus: --min-maf = 0.05. A total of 35 data matrices were constructed to evaluate the effects of different missing data sets on phylogenetic reconstruction.

### 4.4. Phylogenetic Inference

Based on 35 SNP datasets, the VCF file was converted to phylip alignment format using the python script vcf2phylip.py [51]. It should be noted that when there are many gaps or ambiguous sites in the dataset, the -r or --resolve-IUPAC option is needed to resolve heterozygous genotypes to avoid IUPAC ambiguities in the matrices. Then, ModelFinder [52] was implemented to choose the best-fitted nucleotide substitution model for each dataset based on BIC (Bayesian information criterion) values. Then, maximum likelihood phylogenetic trees were reconstructed using IQtree v2.0.3 with 1000 ultra-fast bootstraps and the -bnni option [53,54,55]. The total number of highly supported clades was used as a performance proxy for each independent analysis. To identify consensus and differences among a large set of trees, a density tree was visualized for comparison using the ggdensitree() function in the R package ggtree; Robinson–Foulds (RF) distances between a set of trees were also calculated using the multiRF() function in the R package phytools [56,57,58].

### 4.5. Genetic Diversity and Structure Analysis

Based on a representative SNP dataset, population genetic statistics, including the observed and expected heterozygosity (Obs_Het, Exp_Het), observed and expected homozygosity (Obs_Hom, Exp_Hom), nucleotide diversity (Pi), and inbreeding coefficient (FIS), were assessed using the *populations* program in STACKS [49,50]. The SNP-specific Weir and Cockerham-weighted Fst estimator between two populations was calculated using --weir-fst-pop in VCFtools 0.1.16 [59]. Principal component analysis (PCA) was performed using the option --pca in PLINK v1.9 [60]. Furthermore, admixture analysis was performed to detect the population structure using ADMIXTURE (version: 1.3.0) [61].

### 4.6. Introgression Analyses

To analyze possible conflicting evolutionary signals, the SplitsTree4 software was used to compute an unrooted phylogenetic network for a representative dataset [62]. Splits were created from Hamming distances and visualized as a neighbor net, with each end node representing an individual. To further explore whether tree discordances were due to past introgression, we quantified the Patterson’s D-statistic for all population quartets [63,64]. Calculations were performed using Dsuite Dtrios [65]. In each analysis, the most independent genetic component within the *B. sinensis* species complex was used as the outgroup. For each test, the standard deviation of D was measured from 1000 bootstrap replicates. Then, the observed D was converted to a Z-score measuring the number of standard deviations from zero. To account for multiple testing, we corrected *p*-values with the Benjamini–Hochberg false discovery rate (FDR) [66]. To visualize species/population pairwise comparisons of D-statistic scores and the f4 ratio, two heatmaps were generated using the Ruby script plot_d.rb and plot_f4ratio.rb from https://github.com/millanek/tutorials/ (accessed on 9 January 2023). 

## 5. Conclusions

In this study, by generating RAD sequencing data, we investigated the phylogenetic relationships and population genetic structures of the *B. sinensis* species complex. Our findings indicate that taxa within the *B. sinensis* species complex are indistinguishable phylogenetically and should not be considered as distinct species. In addition, population analysis confirmed that there is a difference in the degree of genetic differentiation between populations. Genetic diversity and differentiation might have been influenced by gene flow. The data generated in this study provide a resource to determine the phylogenetic relationships of ferns in future genetic diversity-related studies.

## Figures and Tables

**Figure 1 plants-13-01987-f001:**
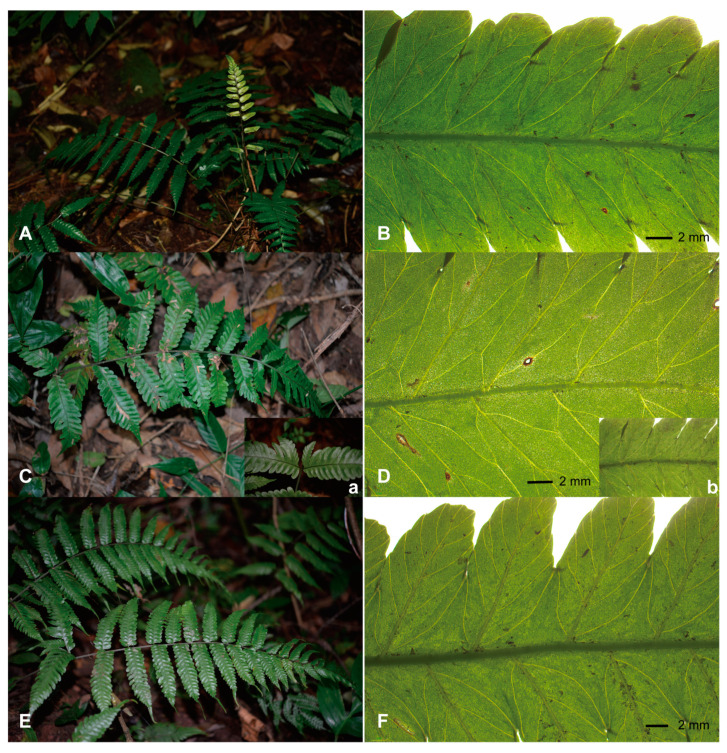
Morphology of the *Bolbitis sinensis* species complex in the wild and variation in its venation pattern. (**A**,**B**) *B. sinensis* and its free-anastomosing veinlets; (**C**,**D**) *B. × multipinna* and multiple anastomosing veinlets; (**E**,**F**) *B. longiaurita* with the basiscopic two or three lobes undeveloped and free-anastomosing veinlets; (**a**,**b**) showing the undeveloped lobes at the base and free veinlets observed in *B. × multipinna*.

**Figure 2 plants-13-01987-f002:**
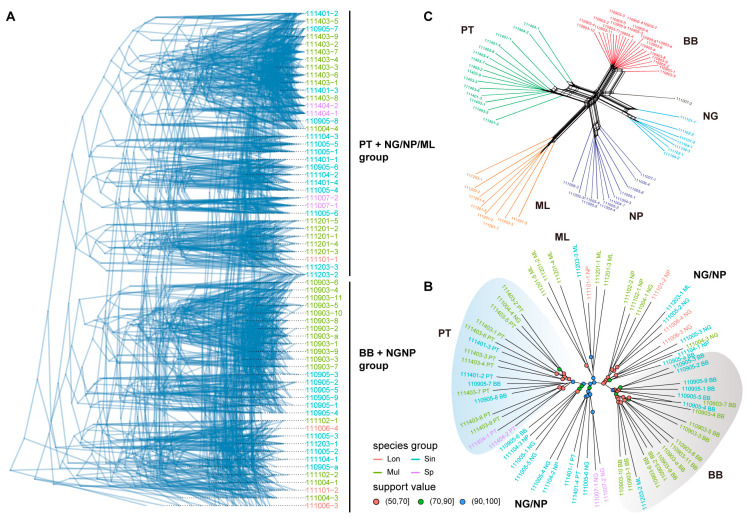
Phylogenetic tree and network analysis of the *B. sinensis* species complex. (**A**) Density tree visualization of 35 maximum likelihood consensus trees. (**B**) Representative maximum likelihood tree with the p1-r0.60 dataset (9467 SNPs) of 65 individuals; the numbers above the branches are bootstrap values based on 1000 replicates, and different colored circles represent those with over 50% bootstrap support. (**C**) Representative neighbor net split network with the p1-r0.60 dataset based on Hamming distances. Branch lengths are proportional to absolute distances calculated from the SNP matrix.

**Figure 3 plants-13-01987-f003:**
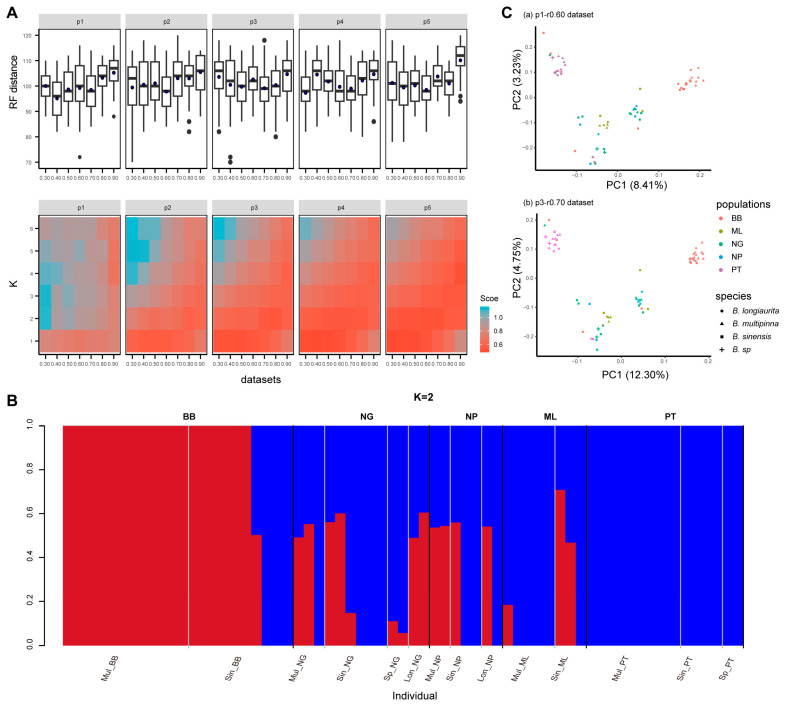
Estimated population structure of the *B. sinensis* species complex. (**A**) The distribution of the RF distance and CV error of structure analysis using 35 SNP datasets; blue points show the average RF distance. (**B**) A structure plot of a representative dataset (p3-r0.70, 1031 SNPs) with K = 2. (**C**) Plot of the first two components of PCA analysis for two representative datasets (p1-r0.60, p3-r0.70). Values in parentheses indicate the percentage of the variance explained.

**Figure 4 plants-13-01987-f004:**
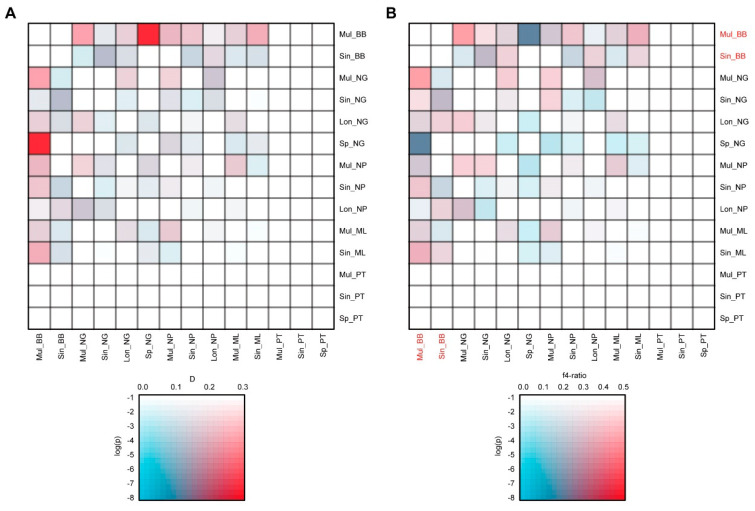
Gene flow is extensive within the *B. sinensis* species complex. (**A**) D statistic values shown between species with significant excess allele sharing. (**B**) Admixture proportions (f4 ratio) between species with evidence of significant excess allele sharing.

**Table 1 plants-13-01987-t001:** Sampling information concerning the *B. sinensis* species complex.

Taxon	Population Code	Locality	Longitude (°)	Latitude (°)	Population Size
*B. sinensis*	Sin_BB	Bubeng, Mengla, Yunnan, China	101.5902	21.6018	10
	Sin_NG	Nangongshan, Mengla, Yunnan, China	101.4313	21.6362	6
	Sin_NP	Nanping, Mengla, Yunnan, China	101.3977	21.6720	3
	Sin_ML	Menglun, Mengla, Yunnan, China	101.3086	21.9080	3
	Sin_PT	Puwen, Jinghong, Yunnan, China	101.0512	22.5769	4
*B. × multipinna*	Mul_BB	Bubeng, Mengla, Yunnan, China	101.5902	21.6018	12
	Mul_NG	Nangongshan, Mengla, Yunnan, China	101.4313	21.6362	3
	Mul_NP	Nanping, Mengla, Yunnan, China	101.3977	21.6720	2
	Mul_ML	Menglun, Mengla, Yunnan, China	101.3086	21.9080	5
	Mul_PT	Puwen, Jinghong, Yunnan, China	101.0512	22.5769	9
*B. longiaurita*	Lon_NG	Nangongshan, Mengla, Yunnan, China	101.4313	21.6362	2
	Lon_NP	Nanping, Mengla, Yunnan, China	101.3977	21.6720	2
*B.* sp.	Sp_NG	Nangongshan, Mengla, Yunnan, China	101.4388	21.6249	2
*B.* sp.	Sp_PT	Puwen, Jinghong, Yunnan, China	101.0512	22.5769	2

**Table 2 plants-13-01987-t002:** Genetic diversity analysis of the *B. sinensis* species complex.

Pop ID	Private	Num_Indv	Obs_Het	Obs_Hom	Exp_Het	Exp_Hom	Pi	Fis
BB	381	17.17512	0.1168	0.8832	0.11173	0.88827	0.11527	0.03708
NG	573	9.8632	0.15594	0.84406	0.14675	0.85325	0.15519	0.03164
NP	221	5.71069	0.16309	0.83691	0.14162	0.85838	0.15601	0.00498
ML	390	6.13749	0.22184	0.77816	0.18258	0.81742	0.20043	−0.01523
PT	616	11.23925	0.16946	0.83054	0.15015	0.84985	0.15775	0.01291

Note: Pop ID, population ID as defined in the text; Private, number of private alleles per population; Num_Indv, mean number of individuals per locus in this population; Obs_Hom, mean expected homozygosity in this population; Exp_Hom, mean expected homozygosity; Obs_Het, mean observed heterozygosity; Exp_Het, mean expected heterozygosity; Pi, nucleotide diversity; Fis, inbreeding coefficient.

**Table 3 plants-13-01987-t003:** Comparison of Fst between different populations.

Weighted Fst	BB	NG	NP	ML	PT
BB	/				
NG	0.0689	/			
NP	0.071	−0.0033	/		
ML	0.0834	0.0258	0.0198	/	
PT	0.1574	0.0678	0.078	0.0817	/

## Data Availability

The datasets generated and analyzed in this article are available in the Figshare repository (https://figshare.com/ (accessed on 28 May 2024)) with DOI: https://doi.org/10.6084/m9.figshare.25906444.v1 (accessed on 28 May 2024). All other data and material analyzed in the current study are included in the manuscript and the Appendix A.

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
