# Peer review of "Relationships Within Bolbitis sinensis Species Complex Using RAD Sequencing"

_plants, 2024, doi:10.3390/plants13141987_

Round 1

Reviewer 1 Report

Comments and Suggestions for Authors

I reviewed the manuscript “Resolving Relationships of Bolbitis sinensis Species Complex using RAD Sequencing” and I think that the study brings good data concerning the species complex focus of the research. In general, the results present good quality and are important to the knowledge of this plant group. I agree with its publication but some aspects should be improved before final acceptance. I have some minor suggestions to the authors:

Title – I suggest to delete the word “resolving” once the relationships were not resolved, the title would be: “Relationships within Bolbitis sinensis Species Complex using RAD Sequencing”.

Key words: I suggest substitute “RAD Sequencing” once it already appears in the title.

Introduction: I have some remarks concerning the Figure 1: when comparing A and B, because of the different zoom used, it is very difficult to see the differences. The same for C and D, and E and F. It would be interesting to use the same scale in both photos.

Results:

Table 1: it is very important to inform the geographic coordinates of the studied populations. Change the Table Header by Sampling information concerning B. sinensis species complex.

Which were the criteria for sampling differences among pops? Only 4 individuals for B. longiaurita and 26 and 31 for the others.

Line 147: monophyletic clade is redundancy.

Figure 2: when presenting bootstrap values, please take into consideration only values above 50, thus, in the corresponding 0-70 should be 50-70.

Line 199: Fst values are presented in Table 3, not in Table 2.

Table 2: in page note: I suggest substitute “in this population” by “per population. Line 209: substitute obsered by observed.

Discussion

From line 273 to line 286, it corresponds to a presentation of the results again, but not discussed. Please compare your data with the literature to elaborate an interesting discussion of these results.

At the end of the discussion section, from line 313 to 320, the authors compare their data with tree species and grasses, but it would be much more important to compare with other ferns, still better with other Polypodiales. I´m not sure if such comparison with Angiosperms make biological sense.

Materials and Methods

Lines 369 and 377: substitute construction and constructed by reconstruction and reconstructed, respectively.

I suggest to perform a multispecies coalescent analysis using a Bayesian approach, this analysis is very useful to species complexes relationships. It would add some interesting evolutionary information.

Author Response

Comments 1: Title – I suggest to delete the word “resolving” once the relationships were not resolved, the title would be: “Relationships within Bolbitis sinensis Species Complex using RAD Sequencing”.

Response 1: Thank you for point this out. We agree with this comment and delete “resolving” from the title.

Comments 2: Key words: I suggest substitute “RAD Sequencing” once it already appears in the title.

Response 2: Thank you for the suggestion, this term has been removed from keywords list.

Comments 3: Introduction: I have some remarks concerning the Figure 1: when comparing A and B, because of the different zoom used, it is very difficult to see the differences. The same for C and D, and E and F. It would be interesting to use the same scale in both photos.

Response 3: Figure 1A/C/E show the overall morphology of the plant, while B/D/F mainly show the diversity of leaf venation, with free-anastomosing to multiple anastomosing veinlets. We added a scale for comparison to the right picture based on morphological data collected from the corresponding specimens. Thank you very much for your suggestion.

Comments 4: Table 1: it is very important to inform the geographic coordinates of the studied populations. Change the Table Header by Sampling information concerning B. sinensis species complex.

Response 4: We have added coordinate information for the population in Table 1, and also made modifications to its headings. Thanks.

Comments 5: Which were the criteria for sampling differences among pops? Only 4 individuals for B. longiaurita and 26 and 31 for the others.

Response 5: Yes, there is a real problem of insufficient sampling of B. longiaurita. We have tried to collect samples of this species, but unfortunately, it is very difficult to find them. It may also indicate that the key taxonomic traits of this species may not be stable traits, making it difficult to find samples with typical traits.

Comments 6: Line 147: monophyletic clade is redundancy.

Response 6: the original sentence has been deleted.

Comments 7: Figure 2: when presenting bootstrap values, please take into consideration only values above 50, thus, in the corresponding 0-70 should be 50-70.

Response 7: Thanks, Figure 2 has been revised, bootstrap value below 50 is not shown in the tree. Additionally, we generated a density tree based on 35 datasets to show the consensus pattern of all topological structures. Besides, considering that “Clade” is more suitable for describing monophyletic branches, but branches in our tree are not monophyletic at all, thus we revised “Clade” to “Group” in Figure 2 and text.

Comments 8: Line 199: Fst values are presented in Table 3, not in Table 2.

Response 8: It has been revised to Table 3. Thanks.

Comments 9: Table 2: in page note: I suggest substitute “in this population” by “per population. Line 209: substitute obsered by observed.

Response 9: The informal phase and wrong spelling have been revised. Thanks.

Comments 10: From line 273 to line 286, it corresponds to a presentation of the results again, but not discussed. Please compare your data with the literature to elaborate an interesting discussion of these results. At the end of the discussion section, from line 313 to 320, the authors compare their data with tree species and grasses, but it would be much more important to compare with other ferns, still better with other Polypodiales. I´m not sure if such comparison with Angiosperms make biological sense.

Response 10: Thank you for this suggestion. In ferns, previous studies on gene flow have primarily focused on hybrids, such as some Alsophila tree ferns. We also found asymmetric parental origins in the hybrids of Bolbitis (unpublished). However, there are currently few studies on asymmetric gene flow for intraspecific population, so we did not find suitable fern examples here.

Comments 11: Lines 369 and 377: substitute construction and constructed by reconstruction and reconstructed, respectively.

Response 11: Those have been revised. Thanks.

Comments 12: I suggest to perform a multispecies coalescent analysis using a Bayesian approach, this analysis is very useful to species complexes relationships. It would add some interesting evolutionary information.

Response 12: Thank you very much for your suggestion. We also used other phylogenetic strategy, such as using varsites as the input or using RAxML to reconstruct the evolutionary tree for entire 35 datasets, but the support value was not as good as those obtained with IQ-TREE. Considering the varied missing rate of our dataset, we think simple method might be more suitable for those datasets with limited informative sites.

Reviewer 2 Report

Comments and Suggestions for Authors

The manuscript “Resolving relationships of Bolbitis sinensis species complex using RAD sequencing” presents a good report about phylogenetic relationship between Bolbitis species. There are a few points that would need review before this manuscript to be recommended for publication:

1 - Introduction

I consider the current introduction presents important background information about the study. However, the last paragraph seems to be more like results/discussion than actual introduction/objective. I do not see a major problem with that approach, but we usually focus on the background and objectives in the introduction. So, I would recommend the authors to use the conventional background/objective approach in the introduction section.

2 - Methods and Results

The sample size of 3 of those populations is concerning me. The problem with a reduced sample size in a small set of your populations is that you might not capture enough variation or homology in the RAD loci data. This could potentially lead to spurious associations between samples.

Moreover, the parameters used by the authors to generate the final set of SNP data also need special attention, especially the ones in the flexible end of the analysis (e.g, -p 1 and -r 0.3).

Imagine the following scenario: If the population of B. longiurata containing 4 samples in total present 2 called genotypes (2 in a total of 4), that is sufficient to satisfy the filter -p 1 and -r 0.3. In this scenario, a substantial portion of the genome may or may not be represented by calls in a few sets of samples in just a single population.

The same rationale applies to -r 0.4, -r 0.5, -r 0.6, etc. Therefore, we can see the -p 1 is one of the most critical filters of this QC step because it creates too much relaxation in the filter stringency.

In these phylogenetic studies, one being able to capture sufficient homology between samples is very important. Therefore, sample size, and SNP coverage are very important aspects. Some of the author's results use the -p 1 and -r 0.6 dataset, which satisfies neither of those sampling strategies (sample size / coverage).

Moreover, the objective of the study is to understand the relationship between samples from Bolbitis sinensis complex. However, in a considerable portion of the results, the emphasis was in the differences between the SNP datasets generated. For a technical point of view, these differences are important, but for this study, I believe it would be more interesting to focus all the analysis in a single dataset (amongst those 35) with sufficient coverage.

3 - Discussion and Conclusion

These sections are fine, but they are based on those flexible results. I recommend the authors to review the results using a more consistent and stringent dataset. If the results remain the same after a more stringent filter, the authors could potentially keep the discussion of these results or make changes to adequate these sections to a possible new scenario.

4 - Summary

In summary, interesting work, but I would recommend applying more stringent filters to methods and results, so the authors could discuss their results more reliably.

Comments on the Quality of English Language

I did not see major issues with the Quality of English. As a good practice, after the revision, I would recommend another pass of review for English language.

Author Response

Comments 1: I consider the current introduction presents important background information about the study. However, the last paragraph seems to be more like results/discussion than actual introduction/objective. I do not see a major problem with that approach, but we usually focus on the background and objectives in the introduction. So, I would recommend the authors to use the conventional background/objective approach in the introduction section.

Response 1: Thank you for your suggestion. We agree with this comment and delete the conclusion sentences in the last paragraph of Introduction section.

Comments 2: The sample size of 3 of those populations is concerning me. The problem with a reduced sample size in a small set of your populations is that you might not capture enough variation or homology in the RAD loci data. This could potentially lead to spurious associations between samples.

Moreover, the parameters used by the authors to generate the final set of SNP data also need special attention, especially the ones in the flexible end of the analysis (e.g, -p 1 and -r 0.3).

Imagine the following scenario: If the population of B. longiurata containing 4 samples in total present 2 called genotypes (2 in a total of 4), that is sufficient to satisfy the filter -p 1 and -r 0.3. In this scenario, a substantial portion of the genome may or may not be represented by calls in a few sets of samples in just a single population.

The same rationale applies to -r 0.4, -r 0.5, -r 0.6, etc. Therefore, we can see the -p 1 is one of the most critical filters of this QC step because it creates too much relaxation in the filter stringency.

In these phylogenetic studies, one being able to capture sufficient homology between samples is very important. Therefore, sample size, and SNP coverage are very important aspects. Some of the author's results use the -p 1 and -r 0.6 dataset, which satisfies neither of those sampling strategies (sample size / coverage).

Moreover, the objective of the study is to understand the relationship between samples from Bolbitis sinensis complex. However, in a considerable portion of the results, the emphasis was in the differences between the SNP datasets generated. For a technical point of view, these differences are important, but for this study, I believe it would be more interesting to focus all the analysis in a single dataset (amongst those 35) with sufficient coverage.

These sections are fine, but they are based on those flexible results. I recommend the authors to review the results using a more consistent and stringent dataset. If the results remain the same after a more stringent filter, the authors could potentially keep the discussion of these results or make changes to adequate these sections to a possible new scenario.

In summary, interesting work, but I would recommend applying more stringent filters to methods and results, so the authors could discuss their results more reliably.

Response 2: Thank you for your thoughtful advice.

I agree with the reviewer that sample size and SNP depth are crucial in population genetics research. Regarding this study, there are some details to share.

(1) The species we studied lacked available nuclear genome due to ferns have extremely large genomes (average >10G) in common, thus we performed RAD sequencing and run without reference pipeline to detect SNPs. This pipeline strategy is to merge all the samples' data to construct a consensus library or a catalog of loci, and then align the population samples to this consensus sequence (like a pesudo-reference genome). While there are some common problems with RAD data, an uneven distribution of SNP sites and a high rate of missing data. It is difficult to apply strict filtering parameters for RAD data (such as requiring that the site be present in at least 90% of individuals for whole genome resequencing data). There have also been many discussions on the validity of RAD data in previously published articles, which can be referred to McKain, et al., 2018. Practical considerations for plant phylogenomics. Appl. Plant Sci. 6.

(2) In this study, we used multiple filtering parameters to try to minimize the impact of missing data. In the data filtering process, we did not assume any species relationships and treated it as a single species. Therefore, for the 5 populations, BB, NG, NP, ML, and PT, the sample size is 22, 13, 7, 8 and 15, respectively. Therefore, the minimum sample size for the smallest group was set to 7, and r was set to 0.3 to ensure that each SNP was present in at least 2 individuals as much as possible. And we hope to uncover a common pattern across different datasets to avoid the heterogeneity of a single dataset.